# A Critical Update of the Classification of Chiari and Chiari-like Malformations

**DOI:** 10.3390/jcm12144626

**Published:** 2023-07-11

**Authors:** Juan Sahuquillo, Dulce Moncho, Alex Ferré, Diego López-Bermeo, Aasma Sahuquillo-Muxi, Maria A. Poca

**Affiliations:** 1Department of Neurosurgery, Vall d’Hebron University Hospital, Passeig Vall d’Hebron 119-129, 08035 Barcelona, Spain; diegofernando.lopez@vallhebron.cat (D.L.-B.); pocama@neurotrauma.net (M.A.P.); 2Neurotraumatology and Neurosurgery Research Unit, Vall d’Hebron Institut de Recerca (VHIR), Vall d’Hebron Barcelona Hospital Campus, Passeig Vall d’Hebron 119-129, 08035 Barcelona, Spain; dulce.moncho@gmail.com (D.M.); alejandro.ferre@vallebron.cat (A.F.); aasmasm@outlook.com (A.S.-M.); 3Department of Surgery, Universitat Autònoma de Barcelona, Bellaterra, 08193 Barcelona, Spain; 4Clinical Neurophysiology Department, Vall d’Hebron Hospital Universitari, Vall d’Hebron Barcelona Hospital Campus, Passeig Vall d’Hebron 119-129, 08035 Barcelona, Spain; 5Sleep Unit, Pneumology Department, Vall d’Hebron Hospital Universitari, Vall d’Hebron Barcelona Hospital Campus, Passeig Vall d’Hebron 119-129, 08035 Barcelona, Spain

**Keywords:** Arnold-Chiari Malformation, Rare diseases, Syringomyelia, Chiari malformation, Classification

## Abstract

Chiari malformations are a group of craniovertebral junction anomalies characterized by the herniation of cerebellar tonsils below the foramen magnum, often accompanied by brainstem descent. The existing classification systems for Chiari malformations have expanded from the original four categories to nine, leading to debates about the need for a more descriptive and etiopathogenic terminology. This review aims to examine the various classification approaches employed and proposes a simplified scheme to differentiate between different types of tonsillar herniations. Furthermore, it explores the most appropriate terminology for acquired herniation of cerebellar tonsils and other secondary Chiari-like malformations. Recent advances in magnetic resonance imaging (MRI) have revealed a higher prevalence and incidence of Chiari malformation Type 1 (CM1) and identified similar cerebellar herniations in individuals unrelated to the classic phenotypes described by Chiari. As we reassess the existing classifications, it becomes crucial to establish a terminology that accurately reflects the diverse presentations and underlying causes of these conditions. This paper contributes to the ongoing discussion by offering insights into the evolving understanding of Chiari malformations and proposing a simplified classification and terminology system to enhance diagnosis and management.

## 1. Introduction

Chiari malformations (CMs) are a heterogeneous group of anomalies of the posterior fossa (PF) and/or the craniovertebral junction (CVJ) characterized by herniation of the cerebellar tonsils below the level of the foramen magnum (FM), often in conjunction with varying degrees of brainstem descent (Figure 1 and Figure 2). These malformations—some authors consider the term ‘deformation’ more appropriate [1,2]—were described in the late 18th century by the Austrian pathologist Hans Chiari from the autopsy of children [3,4]. Since then, this topic has generated controversy and continues to do so. The correct eponym, the phenotypic variations that should be included under the term CMs including new subtypes, the best terminology to define Chiari-like cerebellar tonsillar herniations (TH) found in syndromic and non-syndromic patients, and even the pathophysiology, have been a matter of considerable debate. A review of the literature of the last two decades shows that there are still several problems to be resolved in CM and that each one generates important controversies. Among the most relevant are whether a slight tonsillar ectopia should be considered a ‘normal’ anatomical variant, if incidental patients with CM1 require clinical follow-up, indications for surgery, and, if needed, the most suitable surgical treatment. Professionals, patients, and their families are confused by the variety of opinions and lack of agreement on each topic [5]. 

In the past few decades, the development of MRI has revealed that the prevalence and incidence of CM1 are more common than previously thought, and that Chiari-like cerebellar herniations can be found in children and adults with different diseases unrelated to the four classic phenotypes originally described by Chiari [6,7]. In our review, we propose a simple scheme—modified from previous classifications [1,6]—to differentiate the different types of TH. In addition, we will consider the pathophysiology and suggest the most appropriate terminology to describe patients presenting Chiari-like malformations (Table 2). For most of this review, we have followed the classification suggested by the recently published international consensus documents for diagnosing and treating Chiari malformation and syringomyelia in adults and children [8,9,10].

## 2. A Brief Historical Review of the Chiari Malformation and the ‘Arnold-Chiari’ Eponym

Hans Chiari originally classified CMs into three types (CM1, CM2, and CM3) [4], and later, in 1895, he added a new category (CM4) [3]. Over the last two decades, several subtypes have been added by others (CM0, CM0.5, CM1.5, CM3.5, and CM5), with a total of nine variants described today and summarized in Table 1 [11,12,13,14]. 

Despite various claims to the contrary and many historical studies [11,15,16,17,18], the eponym ‘Arnold-Chiari malformation’ is still used to describe this diverse set of malformations [18]. The main reason for the persistence of this eponym is that any biomedical literature search strategy using PubMed must include the Medical Subject Headings (MeSH) term ‘Arnold-Chiari Malformation’, which was added to MEDLINE in 1966 and remains unchanged since then [19]. The MeSH thesaurus is a hierarchically ordered medical vocabulary maintained by the National Library of Medicine and used for indexing, cataloging, and searching health-related papers on a particular subject, regardless of the terminology used by authors. For the purpose of promoting historical justice [15,16], in this review, we will use the eponym ‘Chiari malformation’ and not ‘Arnold-Chiari malformation’, which, regrettably, is still used by the NLM and Orphanet encyclopedia [20].

In the last two decades, the pathogenesis, screening, genetic inheritance, and surgical treatment of CMs, specifically the Chiari type 1 malformation (CM1) and the frequently associated syringomyelia (Syr), have received increased attention in the medical literature. CM1 is included in the Orphanet database (ORPHAcode: 268882), the reference information source on rare disease research that uses nomenclature (ORPHAcode) that provides a common language in the field of rare diseases [21]. In a PubMed search using the canonical MeSH term ‘Arnold-Chiari Malformation’—conducted on 21 February 2023—we obtained 3925 results, with an exponential increase in the number of studies published since the first paper indexed in 1945 [22] (Figure 3). 

The most common Chiari phenotype, regardless of the TH cutoff used to define it, is CM1. The concept that CM1 was a sporadic condition has persisted for a very long time but has proved to be incorrect. Recent evidence obtained from genetic studies—twin studies, familial clustering, etc.—clearly shows that, although most primary CM1 cases are sporadic, 5–30% of them have a non-Mendelian inheritance [23,24,25]. Non-sporadic CM1 is multigenic and involves several chromosomes—1, 8, 9, 12, 15, and 22—diverse chromosome regions, and many genes [24,25,26]. Urbizu et al., in the study of a Spanish family with multiple affected individuals, discovered the contribution of rare genetic variants in collagen genes to CM1 [27]. Recently, it has been suggested that CMs could be the result of an altered epigenetic regulation of genes involved in the proliferation of brain cells [28].

To understand the modern reclassification of CMs, it is essential to briefly summarize its history and how the eponyms used to describe the different variants have progressed. This framework is necessary to understand how Chiari’s categorization has evolved from its initial description in the late 19th century to the most recent revision incorporated in the documents resulting from the International Consensus Conference held in Milan, Italy, in 2019 (ICC-CM) [8,9,10]. Reading the ‘primary sources’ is the best course of action for updating the history of a certain subject because they include first-hand accounts and research information. The use of secondary sources—translations, evaluations, and interpretations of primary sources—is sometimes misleading and might lead to contradictory and sometimes wrong conclusions. From the 1850s to the early 1900s, when CMs were first described [29], German was the dominant scientific language for the production of scientific knowledge, and with a few exceptions, most early papers dealing with CMs were published in German or French [3,4,30,31]. The authors of this review have read all the primary sources cited. Because of this, some dates and authors listed correspond to those published thus far, while others do not. If the reader encounters any discrepancies, we suggest reviewing the primary sources cited.

To our knowledge, the first description of cerebellar tissue herniating below the FM was reported by Cruveilhier (1791–1874), a French military surgeon and anatomist who, in his atlas of general pathology published in 1829, presented the findings of many cases of spina bifida [30]. In one of the reported cases, the patient had an associated diastematomyelia, and he reported that “*…the upper part of the considerably dilated cervical region, contained both the medulla oblongata and the corresponding part of the cerebellum, which was elongated and covered the fourth ventricle, itself enlarged and elongated*.” [15,30]. Cruveilhier’s work was cited by Chiari and was analog with his description of Chiari type 2 (CM2), where not only the cerebellum but also the brainstem and IV ventricle were displaced into the cervical canal [3,4].

In 1883, the Scottish anatomist John Cleland (1835–1925) published a brief description of an infant with spina bifida and hydrocephalus [32]; some have argued that Cleland was the “*first to publish observations on the morphology of what is termed Arnold-Chiari Malformation (cerebellar ectopy)*…” [33], but after reading the primary sources, we do not believe this to be the case. In his 1883 paper, Cleland reported nine autopsy cases of spina bifida, encephalocele, and anencephaly [32]. In one of the cases with hydrocephalus (Specimen #1, Figure 6), he described that the elongated part of the inferior vermis was misplaced within an elongated IV ventricle [32]. In his paper, Cleland described the cerebellum as having “*…its two lateral lobes completely separated by the vallecula*”, yet he did not mention the ectopia of the cerebellum into the cervical canal [32]. Chiari cited Cleland’s findings, and, regarding his work, he wrote that the cerebellum was described (by Cleland) as “…*two completely separated hemispheres and the nodulus was highly elongated within the elongated 4th ventricle. Unfortunately, this finding is treated only very briefly and the illustration is only macroscopic*.” [3]. Fries et al. reported that Otto Mennicke, a German physician, reported the first case of CM1 in his PhD dissertation [34]. However, Mennicke’s thesis was written in 1891, the same year Chiari published his first report [4]. In addition, Mennicke reported two cases of syringomyelia [35], a disorder that had already been described in 1827 by Ollivier d’Angers [36]. Mennicke gave credit to Chiari for his previous and comprehensive work on syringomyelia [35]. 

In 1894—four years after the first Chiari report—the German pathologist Julius Arnold (1835–1915), working in Heidelberg, Germany, published the case of an infant with spina bifida but without hydrocephalus and described a cone-shaped extension of the cerebellum into the cervical canal. In his own words, “*the cerebellum continues downwards in a ribbon-like mass, broader above, narrower below, and completely covering the 4th ventricle, descending almost to the middle of the cervical cord*” [31]. Arnold did not cite Chiari’s findings [31]. Hans Chiari (1851–1916) was an Austrian pathologist who, while working in Prague, wrote two seminal papers in 1891 and 1895 describing the spectrum of the four malformations included today under the modern eponym ‘Chiari or Arnold-Chiari malformations’ [3,4,37,38] (Figure 4). In his first paper, Chiari presented a preliminary report of the abnormalities found in necropsies of the cerebellum and brainstem induced by congenital hydrocephalus [4,37]. In this 1891 report, Chiari described three different phenotypes and presented the pathological findings in three specific patients [4,37]. The first type was characterized by “*the extension of the tonsils and the medial parts of the inferior lobes of the cerebellum into cone-shaped projections which accompany the medulla oblongata into the spinal canal*” [4,37]. He reported a second type in a six-month-old girl with “*a lumbar myelomeningocele with diastematomyelia, and hydromyelia in the thoracic cord*”, who presented *“…displacement of portions of the cerebellum within the elongated fourth ventricle into the widened spinal canal*” [4,37]. Chiari also described a third type in a five-month-old girl with a cerebellar-cervical hydroencephalocele in whom he found an “…*almost complete displacement of the cerebellum, itself hydrocephalic, into a cervical spina bifida as a result of cerebral hydrocephalus*” [4,37]. Chiari considered that all the anomalies described were a direct consequence of congenital hydrocephalus [4,37] and he expanded his preliminary findings in a second paper published in 1895 [3] in which he added 63 cases of congenital hydrocephalus, of which 14 had CM1 and 7 CM2 [3,37]. In this 1895 paper, Chiari added a fourth phenotype to the first three reported in 1891. Chiari type 4 (CM4) was described in two cases with cerebellar hypoplasia that Chiari believed was also caused by congenital hydrocephalus [3]. In his own words, type 4 cases “*…were distinguished by the fact that they involved developmental disturbances of the cerebellum to be included in the area of hypoplasia, without any displacement of parts of the cerebellum into the vertebral canal*” [3]. 

The widespread eponym ‘Arnold-Chiari’ was introduced in the literature in 1907 by two of Arnold’s students (Schwalbe and Gredig) in a review published in under the title “*On developmental disorders of the cerebellum, brainstem, and cervical cord in spina bifida (Arnold’s and Chiari’s malformation*)” [38,39]. As remarked quite bluntly by Sarnat, “*both authors were former students (i.e., residents) who wanted ‘to honor their professor,’ knowing well that their assertion was fraudulent.*”[40]. For some inexplicable reason, the incorrect eponym is still in use as a legacy of a historical mistake.

## 3. Neuroradiology. Early Radiological Classification of Chiari Malformations

In the early 1980s, the diagnosis of CM1 was usually delayed until neurologic symptoms of the syringomyelic syndrome appeared. In 1983, Paul et al. reported in a series of 71 patients with CM1 who underwent surgical treatment that 56% presented motor weakness and 75% had sensory loss [41]. When these patients were referred to neurologists or neurosurgeons, the diagnostic workup included vertebral angiography, pneumo-encephalography, and conventional or CT-myelography with metrizamide or Myodil [42,43,44]. CM1 diagnosis was usually based on positive contrast myelography. In 1973, O’Connor et al. proposed the diagnosis when patients in the supine position showed the cerebellar tonsils entering the FM in contrast myelography [44]. Before MRI became routine, most neurology and neurosurgery textbooks reproduced Chiari’s original classification with no relevant modifications. In the 1982 edition of the classic Youmans Neurological Surgery—regarded as the definitive reference for neurosurgeons—CMs received little attention. In this edition, the chapter on ‘Anomalies of the Craniovertebral Junction’ used Chiari’s original classification into the three canonical subtypes [45]. Gilroy and Myer’s textbook, a reference for neurologists of around the same date, included two entries dedicated to CMs; the first described the ‘Arnold-Chiari malformation’ (CM2) and the second the ‘Chiari 1 malformation’ [43]. Gilroy and Myer wrote that CM1 “*can be asymptomatic or may cause cerebellar signs and progressive spastic quadriparesis, which can begin at any time during life”* [43]. They suggested that a definite diagnosis should be made when myelography shows “*a typical concave deformity of the columns of dye immediately below the foramen magnum in both the anteroposterior and lateral views*” [43]. 

## 4. MRI: The Breakthrough for the Diagnosis of Chiari Malformations

The history of MRI, which in the late 1980s displaced CT as the primary neuroimaging technique, is strongly related to our current understanding of Chiari malformations. [46]. Since its debut as a diagnostic tool in the mid-1980s, MRI rapidly replaced myelography and became the most widely used neuroimaging technique in neurology and neurosurgery. To our knowledge, the first studies on MRI scanning in patients with CMs were published in the 1980s [42,47,48] and used low-strength magnets (0.15–0.5 T). These studies also revealed that some asymptomatic patients or healthy people may have a mild degree of TH [47,49,50]. The magnet strength of modern MRIs ranges from 1.5 to 3.0 T and, at these field strengths, MRI offers unprecedented detail of the brain and spinal cord in axial, sagittal, and coronal planes with excellent spatial resolution. 

## 5. Classification Schemes: Cerebellar Tonsillar Herniation versus Chiari Malformations

In the last twenty years, new CM phenotypes have been described and pediatric neurosurgeons have reported acquired Chiari-like forms in children with syndromic and non-syndromic craniosynostoses and after ventricular shunting [51,52]. For a better understanding of the etiology of CMs and their proper classification, we will discuss this new information. Several studies have shown that Chiari-like TH is associated with syndromic (Crouzon, Carpenter, Apert syndrome, etc.,) or non-syndromic craniosynostoses and might also develop as a late complication of CSF shunting in children with a ventricular shunt implant [52]. Acquired TH or ‘secondary CMs’ is a term still used in recent reviews [53] and needs to be discussed upfront to differentiate these secondary Chiari-like types from true CMs, in which the primary cause is an embryologically-induced underdeveloped posterior fossa in which TH results postnatally. To clarify if the TH found in various clinical scenarios should be designated as ‘secondary’ or ‘acquired’ CMs, these Chiari-like variations are clinically meaningful and not only of academic interest [7].

Several authors tried to introduce some order into the multiple classification schemes published for developmental, acquired TH, and Chiari-like disorders [1,54]. Buell et al. proposed four mechanisms to explain the pathogenesis of CM1: PF overcrowding caused by underdevelopment of the PF bony structures, increased intracranial pressure, space-occupying lesions of the PF, and downward movement of the cerebellum by events that lower intrathecal pressure such as lumbo-peritoneal shunts [54]. In 2018, Raybaud and Jallo suggested another classification [6], and recently Fiaschi et al., in a comprehensive review of the problem, suggested classifying TH into two different categories: ‘congenital’ (true CM1) and ‘acquired’ (false CM1) [7]. In all proposals, it is widely accepted that TH—usually acute or subacute—induced by intracranial hypertension, space-occupying lesions of the PF, or spinal dural leaks should be completely separated from the group of ‘true’ CMs. 

Here, we will use the terminology shown in Table 2, a minor modification of the scheme proposed by Raybaud and Jallo and Fiaschi et al. [6,7] The following categories will be employed in this proposal: 1. TH associated with high ICP and/or space-occupying lesions; 2. TH induced by craniospinal pressure gradients; 3. Primary Chiari malformations; and 4. Secondary Chiari malformations. The main reason for proposing this modification is that we believe that syndromes with a congenital overgrowth of the brain such as megalencephaly-capillary malformation-polymicrogyria syndrome (MCAP, ORPHAcode: 60040) or Sotos syndrome (OPHAcode: 821) should not be included under the category ‘primary Chiari malformations’ because of their completely distinct pathogenesis.

**1. Tonsillar herniations associated with high ICP or space-occupying lesions.** Acquired TH is a potentially fatal acute or subacute displacement of the cerebellar tonsils through the FM that has been referred to as ‘coning’ in the neurosurgical literature [55]. The TH is detected by neuroimaging and is always the result of increased intracranial volume caused by a variety of pathologies, including intracranial hematomas, posterior fossa space-occupying lesions, traumatic or non-traumatic diffuse brain swelling, or hydrocephalus, among others [55] (Table 2). If the etiological factor is not treated promptly and the patient does not receive respiratory support, acute/subacute TH caused by high intracranial pressure may result in Cushing’s triad (elevated blood pressure, bradycardia, and irregular breathing) and death [56]. In some patients, TH develops slowly, such as in some posterior fossa arachnoid cysts and in some of these cases, there is also an associated Syr [57].

**2. Tonsillar herniations induced by craniospinal pressure gradients.** TH can occur without high ICP but in the presence of chronic spontaneous or iatrogenic spinal CSF leaks (Figure 5). Iatrogenic leaks and TH have been described after insertion of lumbar drainage during or after cranial surgery, lumbar-disc surgery, lumbo-peritoneal shunting, or even after intrathecal pump placement [7,58,59]. When a significant spinal CSF leak occurs, it creates a pressure gradient between the intracranial and spinal compartments that provokes TH, neurological deterioration, or even death if the CSF leak is severe, as occurs in some cases of overdrainage induced by lumbar drainage [60,61]. 

To facilitate brain relaxation and improve tumor exposure during surgery and to prevent postoperative CSF fistulas, lumbar CSF draining is a frequently used technique in skull-base surgery. However, spinal CSF overdrainage has been reported to induce transtentorial and/or cerebral herniations, coma, and motor posturing that revert when the drainage is turned off [62]. Lumboperitoneal shunts (LPS) also frequently induce Chiari-like TH. In children who have undergone LPS, the appearance of Chiari-like TH at varying times after LPS is considered the norm in most infants and children shunted with LPS and traditional differential-pressure valves without gravitational control [63,64,65]. The persistent CSF drainage below the FM level decreases intraspinal pressure, thus creating the driving force for tonsillar displacement. In most patients, the situation is restored by converting the LPS to a ventricular shunt that eliminates the craniospinal pressure gradient and reverts the tonsillar descent [63,65]. 

## 6. Primary Chiari Malformations

The old forms CM1 and CM2, originally described by Hans Chiari, and the newly described phenotypes (CM0 and CM1.5) should be included under the term ‘primary’ CMs, as suggested in the international consensus document [9].CM3, CM4, and CM5 are considered complex unrelated embryological abnormalities and were excluded [8,9] (Table 1). Primary CMs are characterized by the underdevelopment of the occipital bone and a volumetrically decreased PF, which lead to the cerebellum to herniate postnatally downward below the FM and upward throughout the tentorial notch. In 50 CM1 patients, Milhorat et al. demonstrated a reduced supraocciput, an increased tentorium slope, and a short clivus when comparing CM1 patients with age and gender-matched controls [66]. Patients with CM1 may occasionally have various atlas, axis, and/or subaxial cervical vertebral abnormalities. The Klippel-Feil deformity, complete or partial atlanto-occipital assimilation, hypoplasia, dysplasia, or agenesis of the atlas, among others, may be associated with CM1, supporting its relationship with a mesodermal disorder and abnormal segmentation of the somite-derived cervical vertebral bodies [7]. Although we share the opinion held by Raybaud and Jallo that ‘complex Chiari 1 malformations’ should be included under primary CMs [6], due to the lack of agreement regarding these forms and since they were not included in the ICC-CM, here we have segregated them into a different category named ‘Complex Malformations of the Craniovertebral Junction’ (Table 2).

## 7. Secondary Chiari Malformations

A requirement for the correct diagnosis of secondary or acquired CM1 is that these patients have a previous MRI showing normally positioned cerebellar tonsils. In this group, we have included syndromic and non-syndromic craniosynostosis, TH induced by ventricular shunting in children, and the heterogeneous group of excessive growth syndromes such as Sotos and MCAP, among others (Table 2). In secondary or acquired CMs, the pathophysiology is like that of the primary CM forms, except that the PF volume is reduced because of acquired hyperostosis, craniosynostosis, and other forms of bone overgrowth and calvaria remodeling. We have also included in this category disorders that cause cerebellar enlargement but have normal PF volumes, though it is questionable whether these diseases belong in this category.

**Craniosynostosis.** Acquired CM1s is observed in children with craniosynostosis [67]. In a retrospective study of 124 patients with non-syndromic single-suture craniosynostosis explored by brain MRI, Leikola et al. [68] found TH > 5 mm in seven cases (5.6%) [68]. In syndromic or multisuture craniosynostosis, Cinalli et al. reported a CM1 incidence ranging from 50% in Pfeiffer syndrome to 100% in the so-called Kleeblattschädel deformity [51]. In craniosynostosis, the etiologic factor of acquired CM is the premature suture fusion that results in diminished cranial volume and promotes TH [51,67].

**Overdrainage after ventricular shunts.** As already described, lumbo-peritoneal shunts using conventional differential-pressure valves induce TH in most children because of the craniospinal CSF pressure dissociation between both compartments (Table 2). However, TH is also a rare consequence of ventriculoperitoneal shunting or cystoperitoneal shunts in arachnoid cysts [69,70]. In this rare complication, the pathophysiology is not craniospinal dissociation but a slow and progressive thickening of the calvaria and occipital bone and a subsequent reduction in PF volume years after implanting the shunt. Chronic shunt overdrainage causes progressive thickening of the calvaria, microcrania, remodeling of the cranial vault, and a reduction of the whole intracranial volume including the PF [69]. D’Amico et al. reported the case of a 12-year-old girl shunted after resection of a sellar-suprasellar glioneuronal tumor that presented with a 9-mm Chiari-like tonsillar ectopia and drop attacks two years after shunting. The patient improved after standard posterior fossa decompression, C1 laminectomy, and duroplasty [69]. One particularly interesting finding in this report was that at tumor surgery, the PF volume was 173 cm^3^, was reduced to 153 cm^3^ when clinical symptoms presented, and increased to 180 cm^3^ after PF decompression and complete symptomatic relief [69].

**Embryogenic growth dysregulation.** Embryogenic growth dysregulation is a rare disorder that causes the overgrowth of multiple somatic tissues, including the central nervous system. Macrocerebellum is characterized by an abnormally large cerebellum with preservation of its overall shape [71]. As reported by Poretti et al., an enlarged cerebellum is found in syndromic conditions (Costello syndrome, Sotos syndrome, macrocephaly-capillary malformation syndrome, Lhermitte-Duclos syndrome, etc.,) and in isolated non-syndromic conditions [71]. The evaluation of cerebellar architecture in this disorder shows that the cerebellar hemispheres are enlarged secondary to an increase in cerebellar gray matter [71]. As noted by Raybaud and Jallo, “*A developmental excess of neural tissue within a normal-sized posterior fossa may cause a descent of the cerebellar tonsils*” [6]. In macrocephaly-capillary malformation syndrome, around 50% of patients present progressive TH associated with rapid brain growth and progressive crowding of the PF during infancy [72]. In a longitudinal study of 67 patients with macrocephaly-capillary malformation syndrome, Conway et al. found that newborns had a normal-sized cerebellum but developed cerebellar overgrowth and TH during the first year of life with a normal PF volume [72]. Therefore, in these syndromes, cerebellum overgrowth induces a structural mismatch between the skull and the neural tissue facilitating Chiari-like TH. However, the mismatch is because of the increased content and not because of a reduced PF container as in primary CM1.

## 8. Underdevelopment of the Posterior Fossa: The Original Sin in Primary CMs

Primary CM1 and CM2 appear to be multifactorial and arise throughout human embryogenesis, while the exact cause is yet unknown. Three types of evidence indicate a mesodermal insufficiency as the underlying cause of primary CMs: animal models, small-breed dogs with Chiari-like malformations, and morphometric evaluations of the PF conducted on CM patients.


**Experimental models in animals**


Experimental models using vitamin A or trypan blue as teratogens in rodents indicate that the main factor in both CM1 and CM2 pathogenesis is an underdeveloped skull base and small PF as a consequence of disorders occurring during the first eight weeks of human development (human embryogenesis) [54,73]. In humans, the mesoderm develops during the third embryonal week and forms the notochord along the midline and the para-axial mesoderm on either side of it [6]. The mesoderm induces the transformation of the ectoderm into neuroectoderm, which evolves into the neural tube during the fourth embryonal week [6].

The skull and vertebral column are parts of the axial skeleton that develop from the paraxial and lateral plate mesoderm and the neural crest [73]. The paraxial mesoderm forms the somites, which are paraxial mesoderm units that provide the framework for the segmental organization of the vertebral column, trunk musculature, and spinal nerves [74]. Embryologically, the skull is formed by the cranial vault (*neurocranium*), axial skull base (*basicranium*), and facial skeleton (*viscerocranium*) [75,76]. The occipital bone is formed from four somites that generate four pairs of sclerotomes, which will eventually fuse into the occipital bone and be incorporated into the developing axial skull base [6,54,77,78]. In 1981, Marín-Padilla, a Spanish pathologist working at Dartmouth Medical School (Hanover, NH, USA) (Figure 6), hypothesized that the origin of most axial skeletal dysraphic disorders, such as anencephaly, meningomyelocele, and CMs, was essentially a primary paraxial mesoderm insufficiency. This would induce heterogeneous axial skeletal defects depending on whether they occur prior to, during, or after closure of the neural folds [77]. In the case of CMs, Marín-Padilla and Marín-Padilla considered that this mesodermal insufficiency affected embryos *during* (CM2) or *after* (CM1) closure of the neural folds [77]. In his classic experimental work on the golden hamster—using vitamin A as a teratogen—and in human morphometric studies conducted on fetuses, Marín-Padilla classified CM1 and CM2 within a group of disorders that he called ‘axial skeletal-neural dysraphic disorders’ [75,77,79]. In brief, these disorders (i.e., anencephaly, encephalocele, myeloschisis, meningomyelocele, etc.,) share a common denominator despite the vast differences in their neurological anomalies: a mesodermal insufficiency that causes axial skeletal defects [77]. In fetuses of vitamin A-treated hamsters, Marín-Padilla and Marín-Padilla found a shorter and “somewhat elevated” PF in relation to the axis of the vertebral column [77]. In addition, the PF was described as “*short, small, inadequate and funnel-shaped*” (Marín-Padilla and Marín-Padilla, 1981; p. 46) and the cerebellum was displaced posteriorly and inferiorly [77]. Consequently, the neural contents of the posterior cerebral fossa, particularly the cerebellum, “*cannot be properly accommodated and become progressively compressed as they normally grow*” [77].

In modern humans, the basicranium is the interface between the brain and facial structures, and thus is a key player in the development of facial architecture [80]. In his pivotal studies on CM1 embryogenesis, Marín-Padilla demonstrated that all forms of axial skeletal-neural dysraphic disorders also have concomitant abnormalities in the viscerocranium and, consequently, may produce significant secondary changes in the facial skeleton and oropharynx secondary to the adaptation of the facial skeleton to a short basicranium [75,77,79]. This phenotypic structure of the facial skeleton/oropharynx may explain the high prevalence of obstructive sleep-related breathing disorders observed in patients with CM1 [81,82]. A limitation of the paper by Marín-Padilla and Marín-Padilla is that the morphological changes in the occipital bone were consistent in their model; however, in no case did they show significant TH into the cervical canal [77]. This limitation was discussed by the authors and they considered that because the cerebellum develops postnatally, the herniation of the cerebellar tonsils is a late addition of the CMs [77]. According to them, TH “*…requires for its development the rapid postnatal growth of the cerebellum in the presence of a small and inadequate posterior cerebral fossa*” [77]. They tried to keep the affected fetuses alive but the main obstacle was “*the cannibalistic behavior of the hamster’s mother toward her offspring with any kind of malformation however minimal it may be*” [77].

Amin-Hanjani et al., reported two cases of toddlers in whom MRI was performed within six months after delivery because of non-Chiari–related neurological problems and were considered normal [83]. The MRI repeated in the same children at the ages of three years and 18 months, respectively, revealed a CM1 malformation with syringomyelia [83]. This report supports the hypothesis of Marín-Padilla that the bony malformations typically found in CM1 are already present at birth; however, the TH may not occur until cerebellar growth occurs [77]. The cerebellum grows rapidly during the first two years of life and attains nearly two-thirds of its adult weight at the end of the second year [84]. In a large qualitative genomic linkage study, Markunas et al. showed that morphological traits such as PF area, PF height, and clivus length, among other proxies of PF volume, are heritable and strongly related to CM1 [25].

## 9. Chiari-like Malformations in the Cavalier King Charles Spaniel Dog

Comparative pathology is an interdisciplinary study of the biology of human and animal illness that compares disease processes across species to acquire data that may be used for the improvement of human health. A unique opportunity in the understanding of human CM1 has been the detection of an analog of human CM1 in small-breed dogs (Cavalier King Charles spaniel, Griffon Bruxellois, Yorkshire terriers, etc.), which has been named ‘Chiari-like malformation’ [85,86,87].

The term Chiari-like malformation was defined by an international veterinary working group and describes the mismatch between caudal cranial fossa volume and cerebellum leading to herniation of part of the cerebellum and brainstem through the FM [88]. Furthermore, syringomyelia is associated with neurological signs and neuropathic pain in these dogs (Figure 7). The association of occipital bone hypoplasia, overcrowding of the caudal fossa, obstruction of CSF pathways, and syringomyelia was first identified in Cavalier King Charles Spaniels by Rusbridge et al. in 2000 [86]. The high prevalence of Chiari-like malformations in Cavalier King Charles Spaniels is probably the result of breed selection pressure for certain skull shapes and sizes for aesthetic reasons, which has produced dogs with a brachycephalic phenotype [89]. This natural experiment in dogs strongly supports the hypothesis raised in humans of a PF/cerebellum volumetric mismatch as the key factor in primary CMs.

## 10. Morphometric Studies of the Posterior Fossa in Humans

In 1978, Nyland and Krogness, by using the PF ratio methods on plain X-rays, found that both CM1 and CM2 patients had a small PF size [90,91,92]. Three-dimensional (3D) volumetric studies of the PF are plagued by methodological problems because of the software used, its irregular shape, boundary definition, and the heterogenous interslice gaps used in different MRI studies. Therefore, several planimetric (1D) or surface (2D) measures have been used as surrogates of PF volume. Numerous studies conducted on CM1 and CM2 patients using different neuroimaging methods have consistently shown a small PF, short clivus, an increase of the tentorial slope, and a significant reduction in CSF volume in most cases [66,90,93,94,95,96]. 

### Posterior Fossa Boundaries

At birth, the occipital bone is formed by four primary ossification centers joined by a cartilaginous matrix that encircles the brainstem and the upper spinal cord to form the FM [97,98]. These centers are the paired lateral exoccipital segments, the squamous part or supraoccipital, and the anterior basilar portion of the occipital bone (basiocciput) [98]. The supraoccipital is not fused with the condylar portions at birth and it happens around three years of age [98]. The FM can expand and grow postnatally due to its cartilaginous nature.

Most PF measurements used in genetic and neuroradiological studies are derived from the drawing of a five-sided polygon and the additional Chamberlain’s line [25,99]. Recently, Raybaud and Jallo suggested using this pentagon for evaluating midsagittal MRIs of CM1 patients and to define PF geometry [1,6]. In our opinion, the routine use of this method is very useful in daily clinical practice. By drawing these lines, it is possible to raise some flags that allow correct surgical planning and to differentiate isolated forms of CMs from the so-called ‘complex malformations of the craniovertebral junction’, which will be described later. The six lines suggested by Raybaud and Jallo are: 1. Chamberlain’s line; 2. McRae’s line; 3. Clival and Wackenheim line; 4. Incisural line; 5. Tentorial line; and 6. Supraoccipital line. Markunas et al. used all these lines in a genetic linkage study [25], and we used a similar method to validate a supervised machine learning algorithm to detect CM1 and CM1.5 patients with a high risk of sleep-related breathing disorders (SRBD) [100]. 

The PF examination should start by drawing Chamberlain’s line, defined in 1939 as the “base line” to diagnose basilar impression [101]. This line is drawn from the most posterior part of the hard palate and the opisthion (Figure 8). The odontoid tip should be at or a few millimeters below or above this line. Once Chamberlain’s line is drawn in a midsagittal T1-weighted MRI, the five sides of the pentagon are drawn as follows:**McRae’s line (McRL).** McRae’s and McGregor’s lines were described in the late 1940s to assess basilar impression in plain radiographs [102,103]. In 1953, McRae used Chamberlain’s line to evaluate different bony abnormalities in the FM region [103]; however, in a second paper, studying the occipitalization of the atlas, he defined the McRL, which has been extensively used in many studies to describe the position of the normal odontoid peg and/or cerebellar tonsils [102,103]. This line is drawn from the basion to the opisthion and represents the planum of the FM, and its length the anteroposterior width of the FM (Figure 7, line 2). In most studies and clinical practice, TH is evaluated by measuring the distance from the most caudal aspect of the tonsils to a line running perpendicular to the McRL [6,99]. However, since TH is usually asymmetrical, coronal slices are the best method to measure and evaluate it. As indicated by Raybaud and Jallo, in normal individuals, both the McRae and Chamberlain lines are superimposed but diverge when the clivus is hypoplastic [6] (Figure 8).**Clival and Wackenheim lines.** The clival line is the distance between the top of the dorsum sellae and the basion and is a measure of clivus length [6,25]. If the line is extended postero-inferiorly to the upper cervical canal, it forms the Wackenheim line, which is usually tangential to the posterior margin of the odontoid tip (Figure 8, lines 3 and 4).**Incisural line.** This line is drawn from the tip of the dorsum sellae to the union of the vein of Galen with the straight sinus. Its length is equivalent to the antero-posterior length of the tentorial notch (Figure 8, line 5).**Tentorial line.** The line joins the most anterior part of the tentorium to the internal occipital protuberance [6] (Figure 8, line 6).**Supraoccipital line.** This line is drawn from the internal occipital protuberance to the opisthion and represents the length of the supraocciput (Figure 8, line 7).

The full analysis of this model is beyond the scope of this review; however, briefly, the pentagon should be regular in normal subjects but the sides and angles change in CM patients, especially those with CMs and associated abnormalities of the CVJ [6]. In addition, the dissociation between the Chamberlain and McRae lines indicates a short basiocciput and possible basilar impression [6] (Figure 8).

## 11. Do I Have a Chiari Malformation 1? The Origin of the 3–5 mm Magic Rule

This is a common, straightforward question neurologists or neurosurgeons are asked by asymptomatic patients or their families when a cerebellar TH is incidentally found in an MRI performed for a wide variety of reasons (headaches, screening for growth hormone treatment, psychiatric disorders, seizures, etc.). When clinicians face symptomatic patients with significant TH or syringomyelia, the answer is simple. However, they disagree when patients present a moderate TH without symptoms or syringomyelia. In such cases, diagnostic criteria are inconsistent, and patients frequently receive conflicting advice from physicians. Patients with CMs are usually referred to neurosurgeons either by a neurologist, family physician, or the patient themselves when they read in the radiologist’s report that a variable degree of cerebellar ectopia is observed, which is ‘compatible with the diagnosis of CM1′. In the absence of syringomyelia, a rule of thumb for neuroradiologists is to suggest CM1 when the tip of one or both tonsils herniates more than 3–5 mm below McRae’s line [20,47,49,50]. In healthy adults, the cerebellar tonsils are usually located above the level of the foramen magnum (Figure 9); however, CM1 is still defined in most papers and textbooks as a TH into the spinal canal ≥3–5 mm beyond the basion-opisthion line (McRae’s line) [25,27,49,104,105,106].

This widely accepted criterion was used in the most recent Headache Classification (ICDH-III, 2018) to diagnose headaches attributed to CM1 [108], and it was also used in the international consensus conference report on the diagnosis and treatment of Chiari malformation and syringomyelia in adults and children (ICC-CM) [8,9]. In this ICC-CM, the 5-mm rule is a required criterion for CM1 diagnosis. A tonsil descent <3 mm is considered a “physiological variation” (*emphasis added*) and herniation between 3 and 5 mm is a borderline ectopia that deserves follow-up in symptomatic patients [8]. However, upon evaluation, this definition was only agreed upon by 83.7% of an international jury [8]. Therefore, even in a forum of experts, it is obvious that there is no broad consensus and further neuroradiological data are still necessary to reach a 100% agreement on the definition of CM1.

The criteria of normal range, reference interval, or decision limit have been extensively debated in the clinical literature and particularly in interpreting laboratory results [109,110]. Reference intervals define an arbitrary dichotomous interpretation: the value is either normal or abnormal [109]. The conventional approach to defining the ‘normal range’ is using descriptive statistics from a population of ‘healthy’ individuals [109]. In this healthy population, and assuming a Gaussian distribution of the values, identifying the middle 95% of individuals defines the ‘normal range’ [109]. For an in-depth discussion of this topic, we recommend Whyte and Kelly’s paper [109]. When defining normal anatomy, there is increased difficulty. The traditional MRI methodology to describe ‘normality’ is a direct translation of the statistical methodology used to define reference intervals. However, the dichotomy between normal and abnormal includes a third option: an anatomical variant that is clinically irrelevant [111].

O’Connor et al., in 1973, studied the position of the cerebellar tonsils in 100 patients with positive contrast myelography and heterogeneous non-related diseases (cervical spondylosis, migraine, etc.) [44]. They concluded that the inferior tip of the cerebellar tonsils, “*…both of which always lay virtually at the same level, was above the reference line in all cases”* and that the demonstration of the tonsils *“at the level of, or passing through, the foramen magnum is strong evidence of the presence of a Chiari malformation if intracranial tumor has been excluded*” [44]. However, what was abnormal in myelography—tonsils should be above McRae’s line—became normal in MRI in which a tonsil descent of 3–5 mm below this line was accepted as normal or clinically non-relevant. In this section, we will discuss the rationale behind this ‘magic’ number. We will show that, even though this cutoff is firmly ingrained in neurology, neuroradiology, and neurosurgery, the biomedical literature does not produce a single study that justifies this 3–5 mm cutoff for differentiating between normal individuals and CM1 patients. This cutoff is arbitrary, and, in our opinion, a biased interpretation of the first MRI studies published in the 1980s [47,50]. This issue accounts for the inconsistencies found in the CM1 epidemiological studies and merits further discussion. 

The widespread use of MRI in healthy populations and asymptomatic CM1 patients soon raised the question of whether a small degree of cerebellar tonsillar ectopia below this line should be considered ‘normal’. This was based on two key papers written by Aboulezz in 1985 [50] and Barkovich in 1986 [47]. Aboulezz et al. published a retrospective review of 800 MRI exams using a 0.35 T magnet, which included 13 patients with CM1 and CM2 and 82 ‘normal controls’ defined as six normal volunteers and 76 patients with ‘no signs of increased intracranial pressure and posterior fossa abnormalities’ [50]. They reported that the mean vertical distance from the tip of the tonsils to McRae’s line in controls was 2.9 ± 3.4 mm—negative values indicate that tonsils were below the FM level, while positive ones indicate that they were above it. If the data reflected mean ± SD (not specified in the paper), and assuming a Gaussian distribution, in 95% of ‘normal patients’ cerebellar tonsils should not herniate more than 4 mm below the FM. Despite the retrospective nature of this study and the limited number of patients, the authors concluded that “*The diagnosis of Chiari malformation is made when the tonsils extend into the cervical canal for more than 5 mm below the FM*” [50].

In 1986, Barkovich et al. published another paper highlighting that a low degree of TH was of ‘no clinical significance’ in the absence of syringomyelia [47]. This retrospective study included 200 normal patients and 25 with CM1, and measured the position of the cerebellar tonsils below the McRL with 0.5–0.6 T magnets [47]. In patients with symptoms unrelated to CM1, mean TH was 1.0 ± 1.9 mm and no CM1 patient with <3 mm herniations was symptomatic [47]. In an MRI-based morphometric analysis of the PF, Urbitzu et al. included 50 patients scanned with 1.5 T MRI for suspected multiple sclerosis symptoms and compared the morphometric measurements against a cohort of CM1 patients defined by a TH ≥ 5 mm [99]. They showed that a logistic regression model including seven PF measures correctly predicted the probability of CM1 without taking into consideration the degree of TH [99].

The tonsils are paramedian anatomical structures and, in modern MRI machines, slice thickness is usually ~1 mm. In the sagittal T1-weighted images, the evaluator should select the parasagittal section containing the cerebellar tonsil with the greatest caudal descent, and draw the McRL and a second perpendicular line between the McRL and the tip of the tonsil [112,113]. Lawrence et al. studied the variability among seven expert operators in measuring maximum TH in a midsagittal MRI [113]. Interobserver agreement was quite high on average, with an intra-class correlation of 0.83 [113]. However, false negatives—at least one operator reported a descent of <3 mm for TH when other operators measured >5 mm—were documented in 6% of cases, while false positives were observed in 11.4% [113]. Considering the high prevalence of TH in asymptomatic or doubtfully symptomatic patients and the number of symptomatic patients not referred for evaluation because of these classic criteria, the potential error is significant. Barros et al. suggested that the measurement of TH can be significantly improved by standardizing the method and using multiplanar reconstruction of the MRI [112] and their methodology should be considered in the evaluation of doubtful CM1 cases, as well as in research protocols.

## 12. The Canonical Chiari Types and New Phenotypes

The International Classification of Diseases, Tenth Revision, Clinical Modification—more commonly known as ICD-10-CM—is a classification system used for medical reporting in all healthcare settings. In the ICD-10-CM, all CMs are encoded under Q07.0 (Arnold-Chiari syndrome) [114] that contains four codes (Q07.00: Arnold-Chiari syndrome without spina bifida or hydrocephalus; Q07.01: Arnold-Chiari syndrome with spina bifida; Q07.02: Arnold-Chiari syndrome with hydrocephalus; and Q07.03: Arnold-Chiari syndrome with spina bifida and hydrocephalus) [114]. Unfortunately, this classification is very limited and not useful here. 

In this section, we will use the classification proposed by the consensus documents published by Ciaramitaro et al., reflecting the results of the International Experts Jury of the ICC-CM for both adults and children [8,9]. As reported in these documents, there is increasing evidence to support that both CM3 and CM4, as originally described by Chiari, are extremely rare severe embryonic anomalies that are probably not related to CM1 and CM2, so they have been excluded from the classification and are considered of historical interest only [8,9]. Furthermore, evidence accumulated in the last two decades suggests that CMs seem to be a unique malformation with a common underlying developmental disorder but with varying severity in the phenotypic manifestations [106,115]. Here, we will briefly describe the four CM types—CM1 and CM2 as originally described by Chiari and the newly described CM0 and CM1.5 [8]. In Table 1 of this consensus document, it is shown that by using the Delphi method, an agreement was reached of 95.8% for tonsillar herniations secondary to space-occupying lesions, 85.4% for canonical Chiari types, and 81.3% for the newly defined phenotypes (CM0 and CM1.5). Therefore, for this classification, consensus is not synonymous with unanimity but an excellent framework for discussion. 

### 12.1. CM1 (‘Classic CM1′)

Orphanet is a resource dedicated to gathering knowledge and providing high-quality information on rare diseases with the goal of improving the diagnosis, care, and treatment of patients with rare diseases consulted by patients, families, physicians, and researchers [20]. Orphanet defines ‘Arnold-Chiari type I’ (ORPHAcode: 268882) with the traditional definition of “*A central nervous system malformation characterized by caudal displacement of the cerebellar tonsils exceeding 5 mm below the foramen magnum with or without syringomyelia*” [20]. Most papers use the traditional criteria of Aboulezz et al. of 3 mm to define CM1; however, increasing evidence shows that the isolated herniation of the cerebellar tonsils is of limited diagnostic utility and should be considered within a context of clinical and radiological data [66,116]. In the ICC-CM, CM1 is defined as “*a cerebellar dysplasia, included in the wider group of the congenital malformations, due to the abnormal notochordal closure*” [8]. This form corresponds to what Milhorat et al. defined as ‘Classic CM1’ [117]. Pueyrredon et al. reported the histological findings of 43 pediatric patients with CM1in which the tonsils were resected at surgery [118]. Thirty-eight specimens presented histological alterations consisting of Purkinje cell loss, Bergman gliosis, atrophic cerebellar cortex, meningeal fibrosis, and internal granular layer loss [118]. These findings suggested pressure necrosis of the tonsils and not the neuronal disorganization and dysplasias described in patients with CM2.

The symptoms of CM1 depend on whether they present with an associated syringomyelia (ORPHAcode: 3280), which occurs in 45–65% of patients referred to neurosurgeons [66,81,119,120]. The most common symptoms include headache and dizziness [66,81,119,120]. Headaches occur in 64–81% of patients, are predominant at an occipitonuchal location, and in most are exacerbated with Valsalva maneuvers [8,9,10,66,81,119,120]. Neurological syndromes depend on whether patients have syringomyelia associated and, if present, on its severity. Hydrocephalus—defined as an Evans index greater than 0.30 [121]—is found in about 20% of patients [122]. An interesting observation by Ferré et al. is the high prevalence of SRBDs in CM1 [81,82,100]. In a study of 90 patients with CM1 and CM1.5, these authors reported a prevalence of 50% of SRBDs, with a predominant obstructive component [81]. This and other studies [123] confirm that nocturnal respiratory alterations—obstructive hypopneas or apneas and poor sleep efficiency and quality—are frequent in CM1 patients with even minimal symptoms. Consequently, nocturnal polysomnography should be systematically conducted on all patients diagnosed with CM1, especially if they are older than 52, female, or present a short clivus [100].

Morgenstern et al. proposed the introduction of a new CM entity, the so-called Chiari 0.5 (Table 1), to describe patients with TH not exceeding the 5 mm cutoff but having a unilateral or bilateral ventrolateral wrapping of the cerebellar tonsils around the lateral medulla and exiting lower cranial nerves [124]. They presented 41 children in whom MRI showed the tonsils wrapping around the medulla, crossing an arbitrary line bisecting the caudal medulla at the level of the FM, and compressing the medulla [124]. However, most of the patients reported had a mean TH of 6.5 mm [124] and, therefore, in many classifications these patients would be included in the CM1 category. In addition, the ventrolateral position of the tonsils is a frequent finding in patients with both CM1 and CM1.5 and the lateral wrapping of the tonsils is observed in some of the drawings published by Chiari in 1895 (Figure 4). In our opinion, the most important point of Morgenstern et al. is that for MRI evaluation of CM patients, in addition to the conventional midsagittal slices used to evaluate TH, the perimedullary spaces and other cisterns in the PF should be closely evaluated and reported to define if the neural elements are crowded [124].

### 12.2. CM2 and Spina Bifida

Even though it falls outside the scope of this review to explore the clinical distinctions between CM1 and CM2 in detail, we will try to highlight the main differences. CM2 is present at birth and is always associated with spina bifida and brain anomalies, such as polymicrogyria and aqueductal kinking, among others [66]. Spina bifida (most frequently, myelomeningocele—MMC) is a severe congenital malformation of the central nervous system that results from incomplete closure of the neural tube during embryonic development [125]. In the early 1990s, CM2 was the leading cause of death in MMC patients within the first five years of life [126]. MMC is considered the result of a dysgenetic disorder that involves neurulation mechanisms [77] that, according to Marín-Padilla’s hypothesis, can be induced by primary mesodermal insufficiency occurring during the closure of the neural folds [77,79]. In MMC, identified risk factors are maternal diabetes, inadequate folic acid intake, or exposure of the mother to anticonvulsants in early pregnancy, among others [127]. Infants born with MMC require early neurosurgical closure, and they face life-long disabilities, including paralysis below the level of the MMC, bladder and bowel incontinence, musculoskeletal deformities, and spinal cord tethering [127]. In 1942, Russell and Donald presented the pathological analysis of 10 cases of MMC, suggesting that because of the accurate description by Schwalbe and Gredig [39], this form of CM should be called ‘Arnold-Chiari malformation’ [128]. According to the ICC-CM, CM2 (ORPHAcode: 1136) is a “*tonsil herniation associated with spinal dysraphism, such as open or, sometimes, closed, terminal cystoceles*” [8]. 

As Chiari described in his initial reports, brainstem descent is much more severe in these patients than in those with CM1 [3,4]. In 1942, Ogryzlo stated that the main anatomical difference between CM2 and CM1 was “*…the position of the fourth ventricle as governed by the elongation of the brain stem*” [129]. McLone and Dias suggested that spinal dysraphism and intrauterine CSF leaks are the cause of the more severe hindbrain migration into the spinal canal [130] by a mechanism similar to the TH described as a result of craniospinal pressure dissociation (Table 1). In addition, CM2 has associated brain malformations not observed in CM1, such as polymicrogyria, corpus callosum agenesis, a partial or complete absence of the septum pellucidum, grey matter heterotopias, enlarged massa intermedia, and shortened quadrigeminal plate, among others [67]. As MacLone and Dias commented, CM2 could be defined as a ‘pancerebral malformation’ [130]. Hydrocephalus occurs in ~90% of affected patients [130]. The development of the ‘two-hit’ hypothesis was the essential step in the rationale for considering MMC patients candidates for prenatal intervention by open fetal surgery [127]. This theory basically states that two processes are contributing to the spinal cord’s neurological impairment. The first is the failure in neurulation and the second is exposure of the fetal spinal cord to the intrauterine environment [127]. Secondary injury to the spinal cord is a consequence of exposure of the neural placode to the amniotic fluid or toxic components within it (i.e., meconium), which increases the risk of spinal cord damage and long-term sensory and motor deficits, in addition to bowel and bladder as well as orthopedic problems [125,127].

The routine use of high-resolution ultrasound and MRI to detect candidates for maternal-fetal surgery has provided some relevant information regarding CM2 pathogenesis [127]. Kunpalin et al. conducted a systematic review of cranial sonographic findings detected during the second trimester of pregnancy in fetuses with MMC, of which one-third were within the CM2 spectrum [125]. An abnormal PF shape, cerebellum abnormalities, elongated quadrigeminal plate of the midbrain, and a small clivus–occipital bone angle were observed in most MMC fetuses [125]. Ventriculomegaly was detected in 45–89% of cases and most fetuses had a small head due to chronic CSF loss throughout the dysraphic defect [125]. The effect of chronic CSF leak on the severity of the CM2 malformation and small heads has been confirmed by the randomized clinical trial on the Management of Myelomeningocele Study (MOMS), in which the safety and efficacy of prenatal repair of MMC were compared with standard postnatal repair [131]. In this study, the number of infants with any degree of hindbrain herniation was significantly reduced from 96% in the postnatal surgery group to 64% in the infants in whom MMC was closed prenatally [131]. MOMS indirectly supports the concept that a CSF leak at the level of the spinal dysraphism is a relevant etiological factor in the descent of the cerebellum and brainstem found in CM2 patients.

## 13. The Chiari 0 Malformation (CM0)

In 1998, Iskandar et al. described a group of five pediatric patients with syringomyelia without TH who improved after posterior fossa decompression, suggesting a Chiari-like pathophysiology [132] (Table 1). They concluded *that* “*A small posterior fossa could explain the phenomenon that we describe in some of the patients.*” [133] A year later, the same group included an additional patient and conducted PF morphometry, suggesting that the contents of the posterior fossa were compromised in patients with CM0 [133]. 

Kyoshima et al. also reported four additional patients with CM0 and syringomyelia who underwent FM decompression and C-1 laminectomy and in whom a ‘tight cisterna magna’ was the potential underlying cause [134]. Chern et al. retrospectively reviewed 405 surgically treated pediatric CM1 patients, finding 15 (3.7%) with MRI criteria of CM0, all with syringomyelia [135]. An interesting finding already described in the first reported cases was that, despite non-existent TH, the obex was located at or below the level of the McRL in all but two children [135]. For a comprehensive review of this controversial topic, we suggest the review by Bogdanov et al. [136] Moncho et al., in a study of 200 patients aged at least 14 years with CM1—defined as a tonsillar herniation of at least 3 mm below the McRL—found that 14 patients (7%) had CM0 criteria [122].

In the ICC-CM, CM0 is defined as “obliteration of the cisterna magna (due to arachnoid adhesions) and/or volumetrically small posterior fossa, with *cerebellar tonsils positioned at the foramen magnum* (emphasis added) and a syringomyelia in the cervical spinal cord” [8]. However, this definition achieved an agreement of 83.1% [8]. In our opinion, this definition still has room for improvement, as the same document defines a TH < 3 mm as ‘normal’ and CM1 is defined at a cutoff of ≥5 mm [8]. Therefore, by strictly following this classification, it is not clear into which category we should place patients with TH between 3–5 mm with symptoms of CM1 or syringomyelia. Accumulating evidence indicates that CM0, CM1, and CM1.5 are a continuum of CM1, and that removal of the traditional 3–5 mm cutoff could solve most classification problems. This concept was raised by Thompson (Thompson, p. 1655), who wrote: “*Defining CM1 by measurement is not only arbitrary but potentially misleading, risking both over-diagnosis and under-diagnosis*” [2]. The use of robust computer-assisted true volumetric (3D) evaluation of the PF in MRI—instead of the less reliable craniometric (1D) surrogates—in patients with CM1 and CM0 may help to solve this issue. These methods are time-consuming and not easily available yet; nevertheless, preliminary studies suggest that automated segmentation could substitute manual delineation of the PF, therefore improving CM diagnosis [137].

## 14. The Chiari 1.5 Malformation (CM1.5)

In 2004, Tubbs et al. introduced the term CM1.5 to define a phenotype intermediate between CM1 and CM2 but not associated with neural tube defects [138] (Table 1). The main characteristic of this new CM variant is that, in addition to TH, patients also exhibit a caudal descent of the brainstem easily recognized by the obex located below the McRL [138] (Figure 10). The obex is a tiny nervous band located at the inferior part of the foramen of Magendie that joins the two gracile nuclei [139]; it is a midsagittal structure difficult to recognize in the MRI. However, the paramedial nucleus gracilis is easily defined in the MRI and, anatomically, it lies on the same plane as the decussation of the pyramids, which corresponds to the cervicomedullary junction [139,140]. Quisling et al. proposed that the obex/nucleus gracilis (ONG) position is easily recognizable in MRI and is, therefore, a good biomarker to define whether or not the brainstem is displaced inferiorly in reference to the McRL plane [140]. They studied the position of the ONG in anatomic specimens and the MRI of 150 control patients, finding that the ONG was located 10–11.5 mm above the McRL and that the lowest ONG position was 2.5 mm above this line [140]. Oakes et al., in their review of 22 pediatric CM1.5 cases, found that the obex was positioned 9 to 20 mm below the McRL [138].

The true incidence of CM1.5 is difficult to estimate because all centers with a large cohort of patients must retrospectively reclassify them. In a series of 200 CM1 adult patients explored by evoked potentials (52% surgically treated), Moncho et al. found that 24.5% had MRI criteria of CM1.5 [122]. In a pediatric CM1 population (mean age: 11 years), Tubbs et al. observed that 17% had CM1.5 [141]. Apart from the caudal descent of the brainstem, the presenting symptoms are similar in CM1 and CM1.5, without any specific symptoms or neurological abnormalities [14,122,141]. Despite their apparent clinical similarity with CM1, CM1.5 patients have certain distinctions that might change the surgical care plan; hence, it is crucial to classify CMs correctly. Our center maintains a local prospective registry of CM patients evaluated at the Department of Neurosurgery of the Vall d’Hebron University Hospital since 1989. This registry includes a cohort of 1057 adult and pediatric patients who were reclassified in 2013 with the new phenotypes (CM0 and CM1.5) by two of the authors of this paper (JS and MAP). We have 176 patients with CM1.5 (16.7%) in this cohort, of whom 109 were surgically treated. In general, patients with CM1.5 have more retrocurved odontoids and, therefore, greater risk in surgical positioning, as well as more scarring around the tonsils, and require more aggressive management of the cerebellar tonsils (i.e., coagulation or subpial resection of one or both tonsils) than CM1 patients (unpublished results). This finding is in line with Oakes et al., who found that the resolution of syringomyelia in pediatric CM1.5 patients after surgery was significantly lower than in CM1 patients treated with the same surgical technique [138].

Increasing evidence suggests that CM0, CM1, and CM1.5 may represent a continuum of the same disorder in which a crowded PF is the main etiopathogenic factor. The study of Markunas et al. strongly supports this theory, showing clustering of both CM0 and CM1 [142]. In this pivotal study, five families of CM1 patients had at least a first-degree relative with CM0, and the MRI of affected individuals showed similar radiological features to patients with CM1 [142]. Considering that a brainstem descent— defined by a low ONG position in the MRI—is observed both in CM0 and CM1.5 patients, the likelihood that each of the three forms represents variants of the same type of abnormality is very high.

## 15. Complex Malformations: A Puzzle of Mesodermic Disorders

The studies of Creuzet et al. on the avian embryo have shown that in vertebrates, the occipital region of the skull is the only part of the head skeleton to which the mesoderm contributes [143]. The frequent association of CMs and osseous abnormalities of the CVJ suggests a common embryological mesodermal disorder. In complex malformations, the well-known mesodermal insufficiency of the paraxial mesoderm that causes CMs is associated with segmentation anomalies of the occipital and/or first cervical clusters of paraxial mesoderm cells (somites) that flank the neural tube and notochord and are the source of the musculoskeletal system [144]. Each somite further differentiates into sclerotomes, myotomes, and dermatomes, and forms, together with the notochord, the vertebral column [144,145]. Forty-four somites are already present in the human embryo at the end of the fourth week. Although a detailed description of this process is beyond the scope of this review, we recommend that readers refer to the comprehensive review of Menezes for more information [145]. The first four occipital somites (OS1-OS4) and seven cervical somites (CS1-CS7) will form the cervical column, occipital bone, and CVJ [6,145]. The sclerotomes derived from OS1 and OS2 will form the basiocciput, OS3 the exoccipital bone, and OS4 (proatlas),the CVJ [6,145].

The association of CMs with osseous abnormalities of the CVJ has long been recognized, although the prevalence of these anomalies is unknown or inferred from large retrospective studies conducted in centers highly specialized in CVJ disorders [1,6,146,147]. Associated CVJ abnormalities include uni- or bilateral occipital condyle hypoplasia, partial or complete occipitalization of the atlas, anterior or posterior C1 dysraphia, retroflexed odontoid, basilar invagination, platybasia, C2-C3 fusion or Klippel-Feil deformity, and translational or rotational C1-C2 subluxation as the most frequent associations [1,6]. Some of these patients show anterior brainstem compression of the cervicomedullary junction, are at high risk of postoperative neurological deterioration after surgery, and often require transoral/endoscopic surgery for ventral decompression and/or posterior fusion [148]. In a study of 190 patients surgically treated with basilar invagination (BI)—defined as an odontoid tip at least 2.5 mm above Chamberlain’s line—Goel et al. found that 54% had an associated CM [149]. The association of CMs with minor midline bone defects of the posterior arches of C1 or the presence of a moderate retrocurved odontoid are relatively frequent. Bollo et al., in a retrospective review of 101 pediatric patients, coined the term ‘complex Chiari malformations’ to define patients with a high probability of requiring occipitocervical fusion and/or transoral odontoidectomy [150]. They found that three radiological variables defined a subgroup of patients with a high risk of occipitocervical fusion: basilar invagination, the presence of CM1.5, and a clivoaxial angle (CXA) < 125° [148]. The CXA was described in adults by Nagagisma and Kubota in 1983, suggesting that a CXA < 130° in a neutral position may produce ventral impingement of the medulla [151].

In a previous paper, we proposed the term ‘complex CVJ malformation’ to define patients with CMs and at least three of the describer osseous abnormalities of the CVJ [122]. In the last decade, the analysis of cohort of patients included in a prospective registry at our institution’s (Prospective Registry of Patients with Chiari Malformations in Adults and Children; (PROSAC) has revealed that certain conditions in isolation (basilar invagination, CXA < 125°, or complete occipitalization of the atlas) are sufficient to consider any CM a ‘complex CVJ malformation’ that requires occipitocervical fusion and/or odontoidectomy (unpublished results). Although there is no universal agreement on how to categorize these individuals or how to manage them surgically, it is important for the neurosurgeon to have an in-depth understanding of this condition to reduce the risk of isolated PF decompression in these patients. Our department’s approach to managing these patients follows the pivotal paper by Menezes et al., in which reducible or irreducible anterior compression defines the treatment algorithm [152]. To consider a ventral compression irreducible, we use a craneothoracic halo for at least one month before deciding the optimal approach for these patients.

## 16. Syringomyelia in Chiari Malformations: An Elusive Pathophysiology

As stated by Botelho et al., MRI revolutionized CM identification, the study of the disease and its multiple phenotypes being a direct consequence of the post-MRI era and, in particular, of the new high-resolution MRI scans [116]. All CM types are frequently associated with syringomyelia. In the CM1 and CM1.5 clinical series, the frequency of Syr ranges from 30–65% [66,81,122]. That Syr is, in most cases, secondary to masses in the PF which were already described by Langhans in 1891, ten years before Chiari’s first paper [153,154]. Langhans described four cases of syringomyelia, three of which had a tumoral lesion of the cerebellum, and in the fourth, he reported (our translation) “*pressure on the pons and medulla oblongata from above was also present to a high degree, but nothing else was noticeable in the cerebellum at the autopsy, except an apparently very strong development of both tonsils, which projected downwards in the form of two symmetrical pyramidal tumors*” [153]. As noted by Mortazavi et al., this was probably the first description of CM1 and associated syringomyelia [154]. In addition, Langhans introduced the hypothesis that stood the test of time; abnormalities in the cerebral blood flow or CSF dynamics at the FM level were the main etiopathogenic factor of Syr [153,154]. 

In 1985, Logue and Edwards reported 75 patients with communicating syringomyelia—dilatation of the central canal of the spinal cord—and found that in 73 of them (97.3%), the cause was CM1 or a lesion at the FM level (arachnoiditis, cerebellar cyst, etc.) [155]. Most authors agree that blockage of CSF flow around the FM is the main factor in the development of CM1-associated Syr [156]. Robust support for this theory is that space-occupying tumors of the PF may result in syringomyelia [157]. In 2000, Milhorat introduced a useful Syr classification unrelated to intramedullary tumors, distinguishing three types [158]: 1. Communicating syringomyelia or dilatation of the central canal (CC) that is continuous to the fourth ventricle; 2. Noncommunicating syringomyelia or dilatation of the CC that does not communicate with the fourth ventricle; and 3. Extracanalicular syringes or cavities in the spinal cord parenchyma not communicated either with the fourth ventricle or the CC [158]. The study of the pathophysiology of Syr is not within the scope of this review; however, its pathogenesis, and particularly how the CSF enters the spinal cord, is still a matter of debate. In the 1950s, Gardner was the first to propose the hydrodynamic theory in patients with CMs [159,160]. According to Gardner’s theory, the CSF block at the FM generated a “water-hammer” pulse wave during cardiac systole directed at the CC of the spinal cord, which progressively expanded and caused Syr [156,160]. Leung et al. showed, via phase-contrast cine MRI, that CM1 patients had greater cerebellar tonsillar motion than controls, and that this increased motion was significantly reduced following PF surgery to near control levels [161]. 

Whether CSF enters the spinal cord through the opening of the CC in the fourth ventricle, as proposed by Gardner, or from the spinal subarachnoid space through the perivascular spaces, as proposed by Oldfield et al., is still another unresolved issue [162]. The CC is a derivative of the primitive neural tube, which is part of an internal CSF system that includes the cerebral ventricles, the aqueduct of Sylvius, and the fourth ventricle; it is patent in 80% of fetuses and children before the age of nine years and it is obliterated in most adults [163]. Milhorat et al., in the most comprehensive histological study published to date—232 subjects aged from six weeks of gestation to 92 years—found a high degree of CC stenosis [163]. Gardner’s theory was disregarded, probably too quickly, after communication between the fourth ventricle and syringes was not observed in autopsies or surgical reports [162]. The CC is a difficult anatomical structure that lacks detailed studies. Wilson, in 1906, wrote, “*I am not aware of a single text-book figure which adequately illustrates the transitional structural characters met with in passing from central canal to fourth ventricle*” [164]; the citation remains pertinent today. 

The main problem in studying the CC and its entry along the medial sulcus into the fourth ventricle (*apertura canalis centralis, ACC*) is that this submillimetric structure is buried in the inferior triangle of the fourth ventricle, caudal to the area postrema, and is difficult to study in postmortem studies or even under the surgical microscope in telovelar approaches, as it is hidden from the surgeon’s eye by the obex [165]. A step forward was recently made by the endoscopic description of Longatti et al., via flexible endoscopes inserted through the Sylvian aqueduct to explore the fourth ventricle [165]. In their remarkable study, Longatti et al. observed a tiny dimple in the CC in all 44 cases studied [165], where the ACC had a lanceolate or round shape, but none had the gliotic tissue typical of completely closed CC [165].

In the last two decades, the etiopathogeny of Syr formation in CMs has moved from the CC in the floor of the fourth ventricle as the entry point for the CSF to the perivascular spaces of the spinal cord. The increased motion across the FM increases the pulse pressure wave in the spinal subarachnoid space and drives the CSF to the CC through the perivascular spaces, facilitated in part by aquaporin-4 water channels [156]. The most reliable fact, however, is that CM1-associated Syr is significantly decreased or collapsed when surgery successfully reestablishes normal CSF flow around the FM. However, despite an enormous amount of research, how the CSF enters the spinal cord and Syr develops is still uncertain.

## Figures and Tables

**Figure 1 jcm-12-04626-f001:**
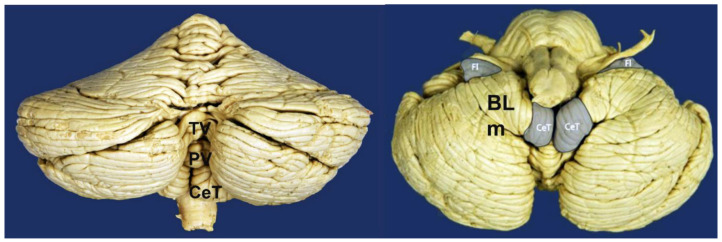
Posterior view (**Left**) and inferior view (**Right**) of the cerebellum and brainstem. **BLm**: Bivental lobe, medial part; **CeT**: Cerebellar tonsils; **Fl:** Flocculus; **PV:** Pyramid of vermis; **TV:** Tuber of vermis. From: Véronique Schotte, https://sites.uclouvain.be/braininteratlas/en/chapter/cerebellum. Images licensed under Creative Commons License 4.0 (https://creativecommons.org/licenses/by-nc/4.0/ (accessed on 7 June 2023) ).

**Figure 2 jcm-12-04626-f002:**
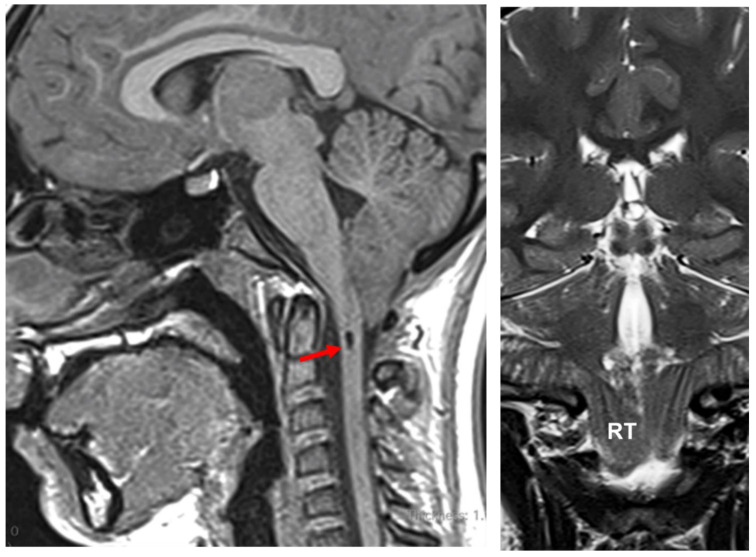
Chiari type 1 malformation. Sagittal T1-weighted MRI (**left**) and T2-weighted coronal (**right**) of a 13-year-old girl referred to neurosurgery with clinical symptoms of dizziness and unstable gait for the last 18 months. Her symptoms progressed in the last three months when she started to report paresthesia in the lower right foot. She did not report headaches. Neurological examination was normal. MRI disclosed a widened foramen magnum, a short subocciput, and asymmetrical herniation of the cerebellar tonsils 19 mm below McRae’s line. The (**right**) tonsil (RT) was slightly more herniated than the (**left**). A small syrinx is observed in the midsagittal T1-weighted slice, just below the lowest tip of the tonsils (**red arrow**).

**Figure 3 jcm-12-04626-f003:**
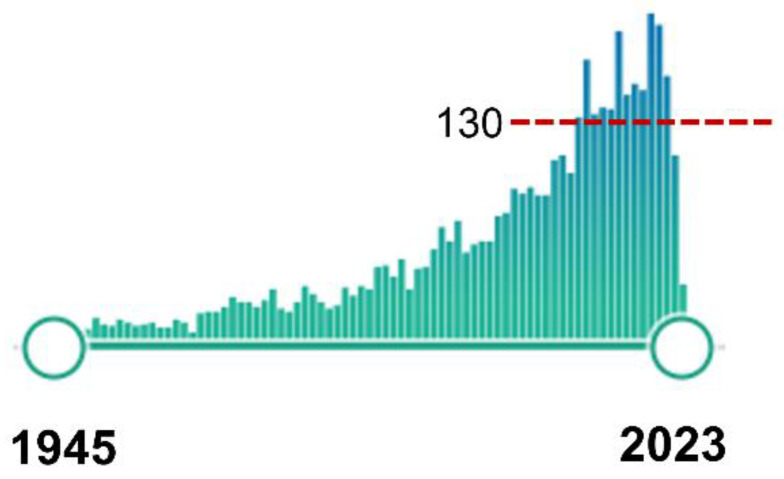
PubMed search conducted using the Medical Subject Headings term ‘Arnold-Chiari Malformation’ on 23 February 2023. MeSH terms are assigned manually by indexers of the National Library of Medicine. The search yielded 3925 records and showed an exponential growth in the number of papers indexed since the late 1960s. In the period 2010–2023, there was a plateau in the number of indexed studies. Since 2010, the average number of papers per year is above 130 (dashed dark-red line). The MeSH term was introduced in PubMed in 1966, when only 17 papers were indexed. Most papers indexed in PubMed are related to CM1. There is general agreement that CM3 and CM4 are rare, severe, embryological disorders that should be included under the Chiari eponym only for historical reasons.

**Figure 4 jcm-12-04626-f004:**
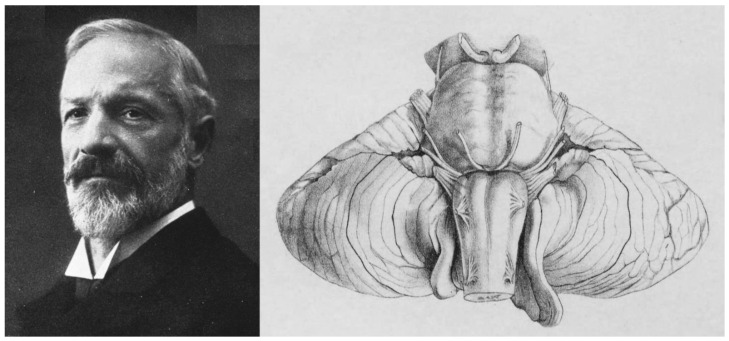
(**Left**): Hans Chiari (1851–1916), an Austrian pathologist who, while working at the University of Prague, described the malformations now known as Chiari malformations. (**Right**): Figure 1 of Chiari’s paper published in 1895 [3] corresponding to the cerebellum of a six-year-old boy with a prior diagnosis of “Hemiplegia spastica”, who died of diphtheria and pneumonia. In his detailed report, Chiari described that the lateral and III ventricles were dilated but the IV ventricle was of normal size. The cerebellum, normally configured, presented a bilateral asymmetrical herniation of the cerebellar tonsils surrounding the medulla oblongata laterally and especially dorsally. The left tonsil ended at the level of the lower edge of the atlas and the right one above the upper edge of the atlas.

**Figure 5 jcm-12-04626-f005:**
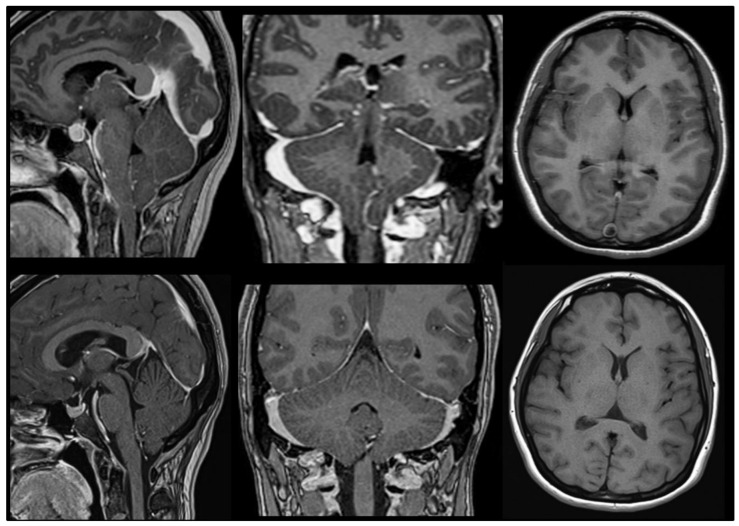
Spontaneous intracranial hypotension and reversible tonsillar herniation after successful epidural blood patches. A 25-year-old female presented with a history of severe orthostatic headaches in the last two years. (**Upper row**): Sagittal gadolinium-enhanced T1-weighted MRI showed intense diffuse contrast enhancement of the pachymeninges, tentorium, and superior sagittal and sigmoid sinus engorgement. The morphology of the PF was normal, and the vetricular size was slightly reduced for the patient’s age. A tonsillar herniation of 14 mm below the McRL is observed. Spinal MRI (not shown) disclosed a CSF leak at the left C7-Th1 level that was managed with three epidural blood patches. (**Lower row**): Complete resolution of the meningeal enhancement, repositioning of the cerebellar tonsils and reversal of the TH eight months after successful epidural blood patches with complete symptomatic relief.

**Figure 6 jcm-12-04626-f006:**
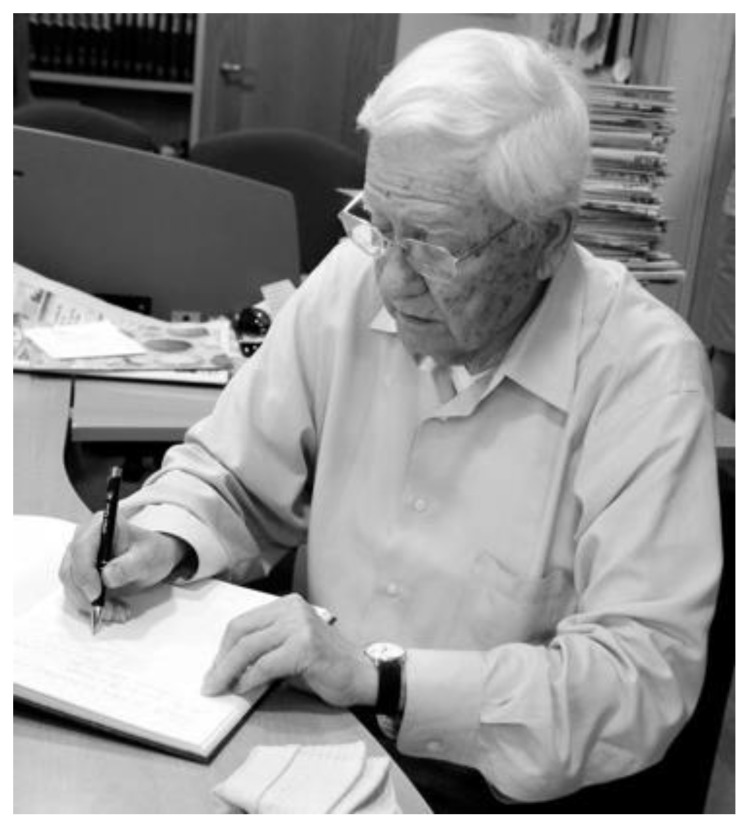
Image of Prof. Miguel Marín-Padilla (1930–2023) in Jumilla, Spain in July 2020 (reproduced with permission of Siete Dias Jumilla). After graduating from the Granada University School of Medicine, Spain, Marín-Padilla emigrated to the U.S. He started his pathology residency at the Mallory Institute of Pathology in Boston, Massachusetts. Marín-Padilla specialized in infants and children and taught general pathology at the Dartmouth Hitchcock Medical Center, Hanover, NH, U.S. In 1981, Marín-Padilla and Marín-Padilla hypothesized paraxial mesodermal insufficiency as the main etiopathogenic factor in CM1 and CM2 and reproduced the axial skeletal abnormalities in hamsters using a single dose of vitamin A as a teratogen [77].

**Figure 7 jcm-12-04626-f007:**
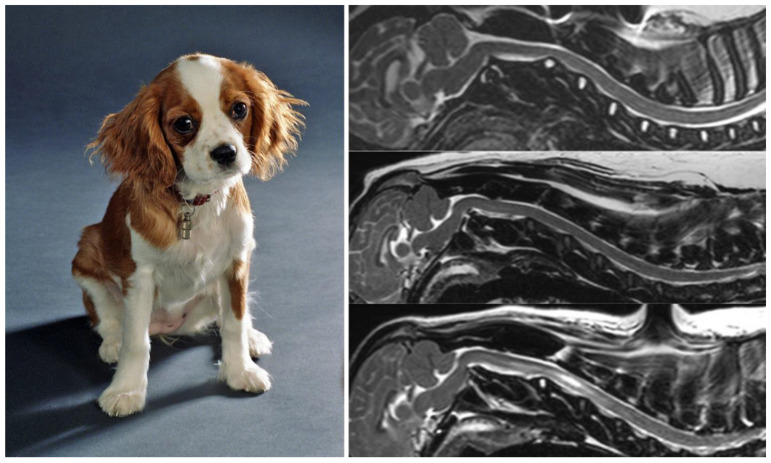
(**Left**): Image of a female Cavalier King Charles Spaniel. T. Voekle. The Cavalier King Charles Spaniel was recognized as a separate breed in 1945: https://rebrand.ly/cavalierdog (accessed on 7 June 2023). Cavaliers descend from the toy spaniels depicted in many 16–18th century paintings by famous artists such as Van Dyck (1599–1561) in *The Three Eldest Children of Charles I*. The Spaniels have flat heads and high-set ears and are a breed in which Chiari-like malformations and syringomyelia are highly prevalent. (**Right**): Reproduced with Permission of Elsevier Ltd., from [89]. In this figure, three sagittal T2-weighted MRI scans of the brain and cervical spine in a flexed position are shown. Top: a brachycephalic dog with a normal cervicomedullary junction; Middle: a Cavalier King Charles Spaniel with Chiari-like malformation; Bottom: a dog of the same breed showing syringomyelia in the cervical spinal cord.

**Figure 8 jcm-12-04626-f008:**
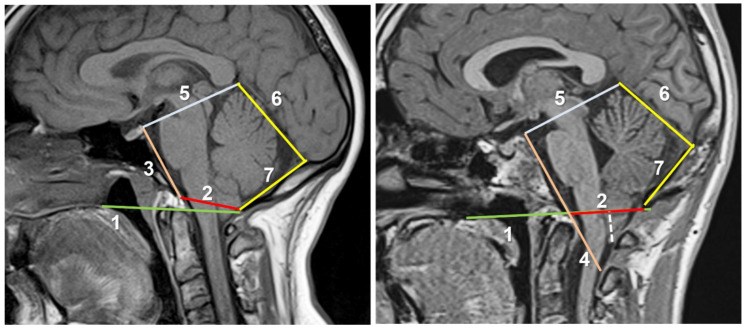
Two adult patients with CM1 (**left**) and CM1.5 (**right**) in whom posterior fossa boundaries have been drawn by using the five-sided polygon and the additional Chamberlain’s line suggested by Raybaud and Jallo [1,6]. These five lines are easily drawn and visually analyzed, and their area has been shown to be a good surrogate for total PF volume and a useful predictor in distinguishing CM1 patients and controls [99]. For a detailed explanation see text. The six lines suggested by Raybaud and Jallo are: 1. Chamberlain’s line; 2. McRae’s line; 3. Clival and 4. Wackenheim line; 5. Incisural line; 6. Tentorial line; and 7 Supraoccipital line. In normal individuals, both the McRae and Chamberlain lines are superimposed (**right**) but diverge when the clivus is hypoplastic (**left**) and it indicates a short basiocciput and possible basilar impression. In most studies and clinical practice, tonsillar herniation is evaluated by measuring the distance from the most caudal aspect of the tonsils to a line running perpendicular to the McRL (dashed line).

**Figure 9 jcm-12-04626-f009:**
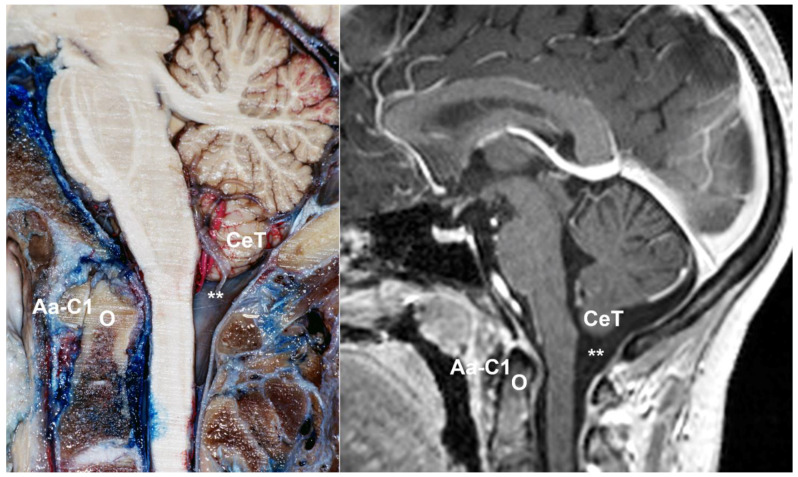
Left: Human anatomical specimen with a midsagittal slice like the midsagittal T1-weighted gadolinium-enhanced MRI shown on the right, corresponding to a 35-year-old female scanned because of headaches. **Aa-C1**: anterior arch of C1; **O**: odontoid; **CeT**: cerebellar tonsil; (******) Cisterna magna. A common problem in assessing the tonsillar herniation (**TH**) in reference to the McRae line (**McRL**) is that midsaggital MRI sections when the slice thickness is >2 mm are prone to error because the medial part of the biventral lobe is difficult to differentiate from the CeT. In doubtful cases, Savy et al. recommend performing coronal MRI, preferably with thin sections, to establish whether any **TH** is present [107].

**Figure 10 jcm-12-04626-f010:**
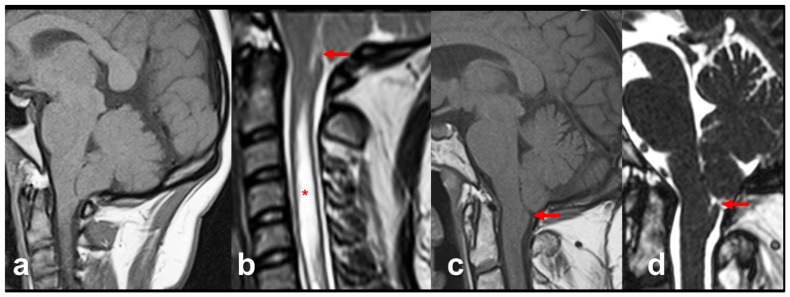
Two patients with CM 1.5. A 27-year-old female (**a,b**). with a clinical history of cough-headaches and and acute severe right cervicobrachial pain associated to patesthesiae of the right hand without motor weakness (**a**). T1W midsaggital MRI showing a 7 mm TH below the McRL (**b**). The obex-gracillis complex is 3 mm below the McRL (red arrow). (*****) A large cervico-thoracic syryngomyelia extending from C2 to the conus medullaris (**c,d**). A 57-year-old female with a clinical history of cough-induced headaches, dizzines and patesthesiae in both hands (**c**). T1W midsagittal MRI showing a TH 4 mm below the McRL. The obex/nucleus gracillis position (red arrows) is 4 mm below the McRL and is better shown in the T2W MRI (**d**).

**Table 1 jcm-12-04626-t001:** A summary of the different types of Chiari malformations (CMs) described in the indexed peer-reviewed literature under the eponym CMs. Due to the exceptionally rare anomalies and the absence of precise embryological investigations, CM3, CM3.5, CM4, and CM5 may be malformations unrelated to the more common CM0, CM1, CM1.5, and CM2. The versions that are kept in the ICC-CM classification are noted in bold and yellow background.

Type	Abbreviation	Author|Year	Main Characteristic
**Chiari 0**	**CM0**	Iskandar et al., 1998 [1]	Posterior fossa crowding. No TH
Chiari 0.5	CM0.5	Morgenstern et al., 2020 [2]	Ventrolateral tonsillar wrapping
**Chiari 1**	**CM1**	Chiari, 1891 [3]	Tonsillar herniation (>3–5 mm)
**Chiari 1.5**	**CM1.5**	Tubbs et al., 2004 [4]	Cerebellar and brainstem herniation
**Chiari 2**	**CM2**	Chiari, 1891 [3]	Spina bifida|Cerebellar-brainstem herniation
Chiari 3	CM3	Chiari, 1891 [3]	Cerebellar-cervical encephalocele
Chiari 3.5	CM3.5	Fisahn et al., 2016 [5]	Cerebellar-cervical encephalocele connected to foregut
Chiari 4	CM4	Chiari, 1895 [6]	Cerebellar hypoplasia or aplasia. No TH
Chiari 5	CM5	Tubbs et al., 2012 [7]	Cerebellar aplasia and occipital lobe herniation

**Table 2 jcm-12-04626-t002:** Etiological classification of tonsillar descent and Chiari-like disorders modified from Raybaud and Jallo [6].

Category	Etiology	Pathologies
**Tonsillar herniation**	Increased volume|High ICP	High ICP Space-occupying lesions in the PF Diffuse brain swelling|Brain edema. Intracranial hematomas Hydrocephalus, arachnoidal cysts
**Tonsillar herniation**	Cranio-spinal gradients	Lumbar drainage Lumbo-peritoneal shunts Chronic spinal CSF leaks
**Primary Chiari malformations**	Mesodermal insuficiency	CM0 CM1 CM1.5 CM2 Complex Chiari malformations
**Complex CVJ malformations**	Embriological mesodermal abnormalities	Basilar impression, platybasia Clivus hypoplasia Retrocurved odontoid Occipitalization of the atlas Klippel-Feil syndrome
**Secondary Chiari-like malformations**	Secondary reduced volume of the skull or neural overgrowth	Syndromic craniosynostosis Non-syndromic craniosynostosis Extrathecal shunting. Overgrowth syndromes (Costelo, Sotos, MCAP, etc.) Non-syndromic macrocerebellum

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
