# Peer review of "A Critical Update of the Classification of Chiari and Chiari-like Malformations"

_jcm, 2023, doi:10.3390/jcm12144626_

Round 1

Reviewer 1 Report

The authors propose a wide review of chiari malformation including historical description, pathophysiology, embryology and then arguments concerning the majors debates of the field.

It is well documented and well written and very interesting even for someone well aware of the fields.

In my opinion, some points are to change for a better understanding of the paper.

First, the manuscrit can be a bit difficult to follow because ideas can be thrown then developed later on. The general plan is confused and i think the quality of the manucript would improve with a bit of dressing (history, embryology, radiology, pathophysiology, discrepancies for example).

I understant that the authors wish to highlight CM1 as a continuous entity mostly related to developmental issues. In my opinion, it is a bit harsh to do so as we all experiment that CM1 is an heterogenous entity with multiple underlying pathophysiology and evidences that supports it. On the other hand, highlighting that cerebellum/posterior fossa mismatch is one of the major cause of CM1 is also important.

Paragraph from lign 129 to lign 145 should be completely rewritten to highlight that CM1 is mostly sporadic but can also carry non medellian inherittance. Saying that CM1 can be associated with genetic disorders (especially congenital bony disorders) for some reason (hydrocephalus, raised venous pressure, OCJ instability, etc ...) instead of CM1 been the result of genetic disorders is much more valid to my opinion.

The proposed classification is made at the wrong place. It should be done at the end of the manuscript after discussion by the authors of all arguments and not at the beginning.

As the aim of the classification is to simplify something that grew more and more complicated, 5 classes is too much. It is hard to understand what is the difference between tonsillar herniation resulting from Posterior fossa tumor and "secondary chiari-like malformation" resulting from small posterior fossa, as in the end it is a secondary chiari resulting from cerebellum/posterior fossa mismatch. Similarly, what is a CM2 except a tonsillar herniation from a CSF leakage in the antenatal period ? On the other hand, including all CM0/1/1.5 in the same "mesodermal insuficency" is incorrect as some of them are resulting from other pathophysiologies (arachnoiditis, instability, ...).

A more precise review of the embryologic pattern of OCJ should help the reader to understand the hypothesis of mesodermal insuficiency. The authors can use the work from Couly and al. DOI: 10.1111/j.1469-7580.2005.00485.x

Author Response

Please see the Word attachment

We would like to express our gratitude to you and the two reviewers for your attention in assessing our manuscript and for your meticulous recommendations. We would like to thank you for this new opportunity to improve our manuscript.

Below you will find our replies to all the reviewer’s remarks and a description of the changes we have made to our manuscript.

Because of the many changes made and the new format of the manuscript, it was confusing to leave the track changes activated in Word. Therefore, we have included a clean Word file and a PDF for comparison between the old and the revised version.

Reviewer 2 Report

 1.       The title is not clear and  doesn’t  match the content.

2.       In abstract: Despite some authors’ skepticism about using eponymic designations and the need to reconfigure the nomenclature toward a more descriptive and etiopathogenic terminology, the increase in variations from the original four categories to nine seems excessive and unjustified. The phrase is very complex and more suitable for literary descriptions than for scientific ones.

3.       Despite the huge and valuable information given in the article, text is poorly structured.

4.       Section: Chiari-like malformations in the Cavalier King Charles Spaniel dog. It would be better if you removed it. the entire article is about diseases in humans with examples of detailed human anatomy. So, why did you switch to animal diseases?

5.       Section: Morphometric studies of the posterior fossa in humans. It would be better if you removed this section or made it part of the introduction. This part is between the sections devoted to classifications of Chiari and Chiari-like malformations.

6.       Section: Do I have a Chiari malformation 1? The origin of the 3-5 mm magic rule . It would be better if you mode this section  part of  the CM0 section.

7.       It is necessary to indicate the classification of Chiari and Chiari-like malformations in the International Classification of Diseases (ICD).

8.       There is no discussion and conclusion section in this paper. It would be better if you could find a pattern between different classifications and make recommendations for the classification of Chiari and Chiari-like malformations based on MRI data and clinical manifestations of the  this disease.

9.       the text between lines 158-173 contains many historical details that could be shortened. these details make it difficult for the reader to concentrate.

10.   In line 209: Arnold did not cite Chiari’s findings [?] but Chiari gave proper credit to Arnold’s contribution. Please add reference link after (findings).

11.   In table No. 1, after each author, reference link must be  indicated

12.   In line 72: To add more fuel to the fire. Scientific style is required in such articles.

Author Response

Dear Editor,

We would like to express our gratitude to you and the two reviewers for your attention in assessing our manuscript and for your meticulous recommendations. We would like to thank you for this new opportunity to improve our manuscript.

In the attached Word file you will find our replies to all the reviewer 2’s remarks and a description of the changes we have made to our manuscript.

Because of the many changes made and the new format of the manuscript, it was confusing to leave the track changes activated in Word. Therefore, we have included a clean Word file and a PDF for comparison between the old and the revised version.

Round 2

Reviewer 2 Report

The topic is relevant and very interesting. In this review, a lot of historical information is devoted to the development of the classification of Chiari formations. The description of the details of the classification is substantiated, taking into account the anatomical and historical features.

In this version, the article is more compact, interconnected and of great interest to readers.

  The topic is relevant and very interesting. In this review, a lot of historical information is devoted to the development of the classification of Chiari formations. The description of the details of the classification is substantiated, taking into account the anatomical and historical features. In this version, the article is more compact, interconnected and of great interest to readers